# Neuronal-Immune Cell Units in Allergic Inflammation in the Nose

**DOI:** 10.3390/ijms23136938

**Published:** 2022-06-22

**Authors:** Vladimir Klimov, Natalia Cherevko, Andrew Klimov, Pavel Novikov

**Affiliations:** 1Immunology & Allergy Dept, Siberian State Medical University, 634041 Tomsk, Russia; chna@0370.ru (N.C.); klimov.lor@mail.ru (A.K.); pavel.n1234@yandex.ru (P.N.); 2Medical Association “Center for Family Medicine”, 634050 Tomsk, Russia

**Keywords:** neuroimmune system, neurotransmitters, neuropeptides, nasal barrier epithelium, neuronal-immune cell unit, single-cell RNA-sequencing, allergic rhinitis, allergic inflammation

## Abstract

Immune cells and immune-derived molecules, endocrine glands and hormones, the nervous system and neuro molecules form the combined tridirectional neuroimmune network, which plays a significant role in the communication pathways and regulation at the level of the whole organism and local levels, in both healthy persons and patients with allergic rhinitis based on an allergic inflammatory process. This review focuses on a new research paradigm devoted to neuronal-immune cell units, which are involved in allergic inflammation in the nose and neuroimmune control of the nasal mucociliary immunologically active epithelial barrier. The categorization, cellular sources of neurotransmitters and neuropeptides, and their prevalent profiles in constituting allergen tolerance maintenance or its breakdown are discussed. Novel data on the functional structure of the nasal epithelium based on a transcriptomic technology, single-cell RNA-sequencing results, are considered in terms of neuroimmune regulation. Notably, the research of pathogenesis and therapy for atopic allergic diseases, including recently identified local forms, from the viewpoint of the tridirectional interaction of the neuroimmune network and discrete neuronal-immune cell units is at the cutting-edge.

## 1. Introduction

Neurons, particularly of the somatosensory and viscerosensory nervous system, endocrine glands, and cells and molecules of the immune system have long been known as the mutual neuroimmune network with a tridirectional communication [1,2,3]. Neuro molecules such as neurotransmitters and neuropeptides, and receptors for them [4], have been at the cutting-edge for a long time, though new generations of neuropeptides are being continuously identified at present. So far, omics technologies have turned into an everyday tool in the research community. A novel scientific paradigm, neuronal-immune cell units [5], have become the subject of discussion in the literature. This review focuses on how neuronal-immune cell units function in allergic inflammation. The prevalence of allergic diseases, especially allergic rhinitis, is a challenge and provides a rationale for researchers’ interest in this pathology. Single-cell RNA-sequencing (scRNA-seq) has enabled a revisal of the landscape of the nasal mucoliated barrier’s cells in the nose in allergic rhinitis [6] and study has commenced of the neuroimmune cross-talk in the unified airway at the modern level [7,8,9,10,11]. Unfortunately, there are still many controversial research results of how single cells and neuro molecules interact in allergic inflammation.

However, the time has come when fundamental data of immunology and neuroscience are being introduced as novel diagnostic and therapeutic approaches into healthcare practice [6]. There are still inefficient medications used in allergic rhinitis, which should be re-studied in terms of the impact on neuronal-immune cell units because this may be a hidden cause behind the low efficacy of some drugs.

## 2. Combined Neuroimmune System

Throughout the entire history of medicine, due to the compartmentalization of medico-biological fields, healthcare specialties and biomedical disciplines long developed in isolation from each other [1,2,3,12]. For example, immunology was separate from neuroscience. On the one hand, alterations within the central nervous system (CNS) such as brain trauma, stroke and spinal cord injury may lead to a predisposition to both exogenous infections and opportunistic microbiota reactivation, which can even be life-threatening, and, conversely, on the other hand, immune attacks are capable of triggering autoimmune neurodegenerative diseases, which illustrates the close connection between the CNS and immune system [12,13]. Using single-cell high-dimensional cytometry, all subpopulations of cells of the immune system are shown to be present in the CNS [2,12,14], and each cell lineage expresses a particular set of neuro molecule receptors and secretes some neuro molecules themselves. So far, strong facts have been accumulated, and it is clear that neurogenic, endocrine and immune factors are produced by the CNS, endocrine glands, and immune system to interact with or even antagonize each other in a tridirectional manner [2,4].

Cells of the immune system, endocrine gland cells, and sensory neurons together share the surveillance over deviations in homeostasis and jointly initiate protective responses if required. The combined neuroimmune system, or “neuroimmune interactome” [15], consists of three counterparts providing tridirectional communication between each other: (1) neuronal, (2) neuroendocrine, and (3) immune [1]. The neuronal part includes neurons in the CNS, spinal cord, and nervous ganglia, and fibers of some types, in particular, somatosensory, viscerosensory, sympathetic, parasympathetic, and enteric fibers, non-neuronal cells, such as neuroglia, neuroendocrine cells (NEC), and various neuro molecules (see Table 1). In the 1930s, H. Selye described the hypothalamic–pituitary–adrenal (HPA) axis [16], which can be currently considered as the second counterpart of the mutual neuroimmune system. Notably, the hormonal effects to a large extent last longer than those of cytokines and most neuromediators. Genes in cancer progression and metastasis, including genes responsible for neuroendocrine control (e.g., the receptor synthesis for neurotransmitter, insulin, progesterone and other hormones) and appropriate epigenetic modifications are found to have a predisposition to such severe pathologies as various organ-specific forms of cancer [4]. NEC, which may be related to both neuronal and neuroendocrine compartments, are significant for the communication between the barrier epithelium and CNS and can sense microbial products [17]. Consequently, the immune system represents the third counterpart of the combined neuroimmune system.

## 3. Neuro Molecules of the Neuroimmune Network

Neuro molecules, hormones and immune cell-derived molecules play regulatory and effector roles at the systemic and local levels to engage all body resources to (1) fight against infectious invaders or cancerous cells wherever they appear, and (2) provide allergen tolerance if environmental allergens enter the body. Therefore, depending on the circumstances, the combined neuroimmune system can trigger either active adaptive responses or tolerogenic mechanisms [26]. However, these processes can be beneficial or harmful. Neurotransmitters, such as L-glutamate (L-Glu) and dopamine (DA), can exert exitotoxic and neurotoxic effects [27,28,29], whereas pro-inflammatory cytokines such as IL-1, IL-6, and TNF-α are able to evoke a “cytokine storm” and destroy the body functioning up to a life-threatening condition, e.g., in COVID-19 [30,31]. In addition, the neuroimmune network is closely linked with the concept of “neurogenic” inflammation [21] since nociceptive primary afferent neurons express many neuro molecules displaying influence over immune cells, enable damage to the tissues of physiological barriers that they innervate, and modify the sensory input before it reaches the CNS.

Including neuro molecules, which exercise influence over target cells, there is a whole family of particular growth factors, neurotrophins, which originate from the immune system, unified airway, skin, and CNS. They are responsible for many processes, such as nociceptive transmission, survival of neurons, immune regulation, and even participation in allergic and “neurogenic” inflammation [18,32]. 

Nowadays, no more than 12 small-sized molecules of neurotransmitters and over 100 neuropeptides have been identified [33]. This review focuses on some of these molecules only. An array of neuro molecules may be divided into conventional (excitatory or inhibitory) neurotransmitters, neuropeptides, neurohormones (or peptide hormones), atypical neurotransmitters, and non-classified neurotransmitters. In contrast with conventional neuromediators, atypical molecules are released into the synaptic cleft from the postsynaptic membrane. All the mentioned neuro molecules serve as an instrument with which the combined neuroimmune system functions at the systemic level and regional level and contributes to allergic and “neurogenic” inflammation [18,21,22]. Neuro molecules display multi-directional effects and distinct forms of interaction with immune cells that can be controversial or counteractive. For example, almost every neuromediator may exert both protolerogenic and pro-immunogenic activity, but, as a rule, only one activity type may be considered as predominant. For simplicity, since neuro molecule classifications based upon identical or shared biological properties and sequence homology are complex and problematic, we structured all these molecules into pro-immunogenic and protolerogenic profiles in a logically grouped manner and according to references, taking into account their preferable participation in upregulation or downregulation of allergen tolerance (see Table 2) [26].

Receptors, which are required for binding neuro molecules released from producer neurons or other cells to get downstream effects in the target immune cells, are divided into (1) ionotropic and (2) metabotropic, where ionotropic receptors have been historically subdivided into (i) ligand-gated (so far, it is the prevalent subtype), (ii) voltage-gated, and (iii) stretch-activated subtypes. In fact, the ionotropic receptors function as ion channels in a direct manner, being permeable to sodium, potassium, calcium, chloride, and bicarbonate ions. In contrast, metabotropic receptors, or transmembrane G-protein-coupled (GPCRs) receptors, involve second messengers and various signaling pathways inside the cell acting on target cells indirectly [98,99]. Most neuropeptides use metabotropic-like GPCRs as well as some neurotransmitters exploit. In addition, GPCRs are currently the medication targets for many approved drugs, and pharmacology approaches serve as an important instrument for the study of GPCRs [99]. Particular molecules, *transporters*, enable the movement of neurotransmitter and neuropeptide molecules through membranes and inside the target cells (see Figure 1). Meanwhile, two neurons interact with each other using synaptic transmission neuro molecules and receptors frequently located on the postsynaptic membrane of a receiving neuron. These are the same molecules which were mentioned above. There are as many neuro molecule ligands as receptors for them, and new molecules are being continuously identified; however, neuro molecule receptors are not the main subject of this review, therefore we are confined to a brief description of them.

## 4. Functional Structure of the Muciliary Barrier in the Nasal Cavity 

Nowadays, omics technologies [100] such as scRNA-seq, mass spectrometry (MS)-based proteomic methods such as matrix-assisted laser desorption/ionization time-of-flight MS (MALDI-TOF/TOF-MS) [101] and liquid chromatography tandem MS (LC-MS/MS) [102], and phenotyping could elucidate the heterogeneity of cells and mucus proteins [103] as new biomarkers, neuropeptides, and enzymes critical for neurotransmitter synthesis, which are involved in neuronal-immune cell units at the mucociliaryneuroimmunologically active barrier between inhaled air and unified airway [7,8,9,10,11]. However, research of neuro molecules in allergic rhinitis by proteomic and metabolomic techniques is still in progress. The nasal epithelium is richly innervated with olfactory, nociceptive sensory nerve fibers, as well as sympathetic and parasympathetic fibers [6]. The olfactory nerve fibers inside the cranial nerve I provide afferent innervation linked with the sensation of smell. The endings of nociceptive sensory nerves inside a cranial nerve V’ branch sense various stimuli: mechanical, chemical, hypertonic saline, cigarette smoke, thermal, aero-allergenic, etc. The sensory nerve ending secrete SP, NMU, VIP, and CGRP [6,104]. The parasympathetic system controls the nasal secretion and produces ACh, which controls mucus releasing, and VIP that stimulates the local vasodilatation. The sympathetic system regulates the nasal blood flow due to the secretion of NE [101]. Therefore, the most important neurotransmitters and neuropeptides for the nose are ACh, NE, SP, NMU, VIP, and CGRP [6,104,105].

The barrier epithelium contributes to regional innate and adaptive immunity, and neuroimmune interaction between cells by which it is formed. Recently, the use of a transcriptomic technology, scRNA-seq, enabled a revisal of the contemporary structure of barrier epithelium and identified new cell lineages present among canonical cell types [7,8,9,10,11,106]. The second type of the nasal epithelium is the olfactory epithelium. However, the olfactory epithelium is outside of the scope of this review. Submucosa, or the subepithelial region, is rich in immune cells, which develop allergic inflammation and constitute the sites of the chronic inflammatory process also regulated by neuroimmune-derived factors (see Figure 2) [26].

The landscape of the nasal ciliated pseudostratified columnar epithelium is composed of prevalent and rare cell lineages, among which multiciliated epitheliocytes, goblet and basal cells are predominant. Mature mucociliated epithelial cells are crucial for innate and adaptive immunity due to their ability to (1) build barriers against any environmental foes; (2) implement the effective back mucociliary clearance directed from the nasopharynx outside the nose; (3) produce alarmins, IL-25, IL-33, and thymic stromal lymphopoietin (TSLP), as a danger signal for group 2 innate lymphoid (ILC2) cells, allergen-presenting dendritic (APDC) cells, and type 2 helper T (Th2) cells; (4) secrete antimicrobial peptides such as β-defensins and cathelicidins [107]; (5) express Toll-like receptors (TLR) and other pattern recognition receptors required for the initiation of immune responses to microbial antigens; and (6) synthesize neurotransmitters and neuropeptides contained in the epithelium nerve endings, NEC, and epitheliocytes themselves [7,26]. In addition, apically disposed tight junctions between epitheliocytes control ion and metabolite transport, while the adherens junction along with desmosomes formed by cell adhesion molecules such as cadherins provide the epithelium integrity. Destroyed epithelial integrity facilitates the entry of environmental microbes and allergens to the body [10,105,108]. Undoubtedly, mucociliated epitheliocytes of the barrier epithelium belong to the neuroimmune system as an essential compartment. Precursors to these cells are so-called deuterosomal cells (see Figure 2).

The secretory nasal epithelium’s cells, goblet and club cells, produce mucus and antimicrobial peptides contributing to the first frontline of innate immunity. Goblet cells are divided into two subsets, among which “goblet 2” cells are more expressed in the nasal epitheium than lower airways and are significantly involved in neuroimmune processes regulating cell motility, differentiation and sensory perception [7]. Paradoxically, the main protolerogenic neurotransmitter GABA stimulates goblet cell proliferation and mucosa remodeling in respiratory allergies [74]. It has been reported that goblet cells can act as precursors to multiciliated cells [9]. Another secretory cell type, club or Clara cells, is less abundant in the nose and is more prevalent in the lower airways [9].

Basal cells are multipotent stem cells that give rise to secretory cells and mucociliated epitheliocytes during homeostatic maintenance of the epithelial barrier and during regeneration if injury occurs. Suprabasal cells represent an intermediate stage between basal and club cells [7,10]. Tuft cells or “brush” cells, close to the enteroendocrine and even NEC, exhibit chemosensory activity and are able to release GCRP, SP, IL-25, and cysteinyl leukotrienes [109]. The expansion of tuft cells in the upper airways occurs after the inhalation of common aero-allergens via activation of the cysteinyl leukotriene synthesis linked with the promotion of the allergic inflammation pathway.

The first barrier’s frontline in contact with the environmental space is the nasal cavity, a part of the upper respiratory tract.

## 5. Allergic Inflammation at the Sites of Nasal Mucosa 

The prevalence of well-known and new pathogenic types of allergies in the unified airway, including the nose, is increasing. Allergic rhinitis has become a global healthcare challenge frequently linked with allergic asthma that impacts upon the quality of life, social and educational activity, and results in an increased financial burden on healthcare [110,111,112]. In recent years, new forms of respiratory atopic disorders, such as local allergic rhinitis, local allergic asthma, “dual” allergic rhinitis, and local allergic conjunctivitis, have been described [113,114,115,116,117]. Although local allergic rhinitis was described approximately a decade ago, its detailed immunopathogenesis is not understood yet. 

In clinical immunology, allergic rhinitis is considered as an atopic condition caused by Th2-mediated sensitization to aero-allergens, including house dust mites, oversecretion of IgE antibodies, development of chronic allergic inflammation in the nose, and involvement of neuroimmune regulation of the nasal epithelium and subepithelial region in pathological circumstances [26]. Any atopic disease manifests only in predisposed individuals prone to respond to environmental allergens in a selected manner, whereas allergens are not dangerous for most people [118]. Many cells and biomolecules take part in allergic inflammation in the nose (see Figure 2). The neuroimmune network defines the maintenance or breakdown of allergen tolerance in the upper and lower airways, therefore its study could become key to unveiling unknown secrets of conventional and new local respiratory diseases. 

Unfortunately, there are only a few publications on nociceptive sensory neuron-derived peptides and their effects in the patients who suffer from allergic rhinitis. Although the neuronal functional structure in the nasal epithelium is well described, whether these neurons respond to allergenic molecules in a direct or indirect manner requires further investigation [6]. 

## 6. Neuronal-Immune Cell Interplays in the Upper Respiratory Tract 

Atopic allergic disorders based on chronic allergic inflammation are those cases, which, in contrast to adaptive immune responses, require allergen tolerance re-induction to counteract pathological processes and alleviate clinical signs and symptoms [25,26]. The newest area of research related to the “neuroimmune interactome” is the study of discrete neuron-immune cell units in healthy and allergic conditions that taken together or considered as autonomic can unveil subtler mechanisms and pick up unforeseen facts of neuroimmune cross-talk, create novel biomarkers, and contribute to potential therapeutic approaches. The neuroimmune system in total and separate neurotransmitters, and neuropeptides, followed by neuronal-immune cell units, are becoming a novel research paradigm [5].

### 6.1. Neuronal-Lymphocyte Unit

T cells and B cells represent the main cell lineage in all immune processes related to both innate and adaptive immunity due to how they can learn and be taught, to sense and be tuned to recognizing “non-self” and “former-self”, to be active or silent, to be faithful to the organism and recombine their genes to respond to invaders better, to memorize enemies or, sometimes, if cancer is in progress, to harm the body [26].

According to in vitro research, T cells and B cells are modulated by NE via β_2_AR signaling [119,120] and by ACh through a7nAchR signal transduction inside neurons of vagus-splenic synapse, which is the uncommon cholinergic anti-inflammatory pathway [2]. The neuronal-lymphocyte cross-talk with the participation of the vegetative nervous system-derived neurotransmitters proceeds in a contradictory manner: (1) T cells polarize into Th2 and type 17 helper T (Th17) cell pathways, and, interestingly, differentiate into peripheral regulatory T (pTreg) cells, while B cells mature into plasma cells, which secrete antibodies; (2) T cells do not develop by the type 1 helper T (Th1) pathway, inflammasome activity is inhibited, and pro-inflammatory cytokines such as TNF-α are not synthesized [5,27].

DA via D_1_ and D_4_ receptors promotes Th2 and Th17 cell differentiation, whereas through D_3_ upregulates naïve CD8+T cells migration, but via D_2_ DA exerts protolerogenic properties [2]. Using both iGluR and mGluR receptors, L-Glu stimulates Th1-dependent adaptive immune responses of T cells and their migration [2,27]. Furthermore, histamine is a pro-inflammatory multipotent mediator of allergic inflammation; this neurotransmitter via H_2_ upregulates the Th2 response and inhibits pTreg cells [42,45]. Depending on environmental factors, SP amplifies the polarization of Th1 and Th17 mediated responses and production of pro-inflammatory cytokines and chemokines by CD4+T cells [2,27,55]. Conversely, the ability of SP to induce a Th2-mediated response of T and B cells and allergic inflammation has been described [13]. Through the NMUR1 receptor, NMU stimulates Th2 cell migration to the inflammatory sites launched by mast cell degranulation and amplified by eosinophil recruitment [59,62].

T cells can synthesize serotonin, whereas B cells enable taking it up. Preferably, serotonin in vivo influences lymphocytes as an inhibitor of their activation and migration [67]. Effects of CGRP in relation to the Th2 pathway are controversial [17,82,83]. Protolerogenic neurotransmitters, GABA and glycine, and neurohormone oxytocin downregulate the clonal expansion and differentiation of T cells and B cells suppressing the inflammatory process [27,75,78,82,83]. At the same time, NMU, SP, and VIP are reported to display an opposite capacity of promoting Th2-mediated response and allergic inflammation [6,17,22].

Thus, within the neuroimmune network, lymphocytes exert a functional plasticity [2,22], depending on pathologic circumstances and irrespective of typical predominant protolerogenic or pro-immunogenic profile of neuro molecules.

### 6.2. Neuronal-Dendritic Cellunit

The DC lineage is the most important cell type for all adaptive response pathways, allergen endocytosis, processing, allergen uploading on class II HLA molecules, and presenting allergen/HLA II to lymphocytes. Furthermore, DCs are characterized by high heterogeneity, multiple subsets, and distinct stages of maturation at which they operate in a different manner [121]. 

Notwithstanding long-term research, neuronal-DC communication is still a topic of discussion [17]. First of all, there are some antagonistic subsets of DCs, e.g., allergen-presenting dendritic (APDCs) cells and tolerogenic dendritic (TDC) cells, which express receptors to various neuro molecules and exhibit different downstream effects. In a model of airway inflammation [3], the functioning of APDCs upon Th2-dependent responses to allergens entering the respiratory tract, but not upon Th1-mediated inflammation, is upregulated by CGRP, a preferable protolerogenic neuropeptide [13,26]. Catecholamines, including NE, regulate DC through β_2_AR signaling, suppressing Th1-dependent pathways and promoting [122] or, conversely, inhibiting Th2 and Th17 responses [63]. Neuropeptide VIP related to the parasympathetic system [13] promotes TDC as well as pTreg in an indirect manner [5,17].CCR7 expression critical for DC migration throughout the body is controlled by serotonin via 5-HT_7_ receptors [67,123]. Unfortunately, due to contradictory and fragmentary data in both mice and humans, there is not yet full and clear understanding of the neuronal-DC unit.

### 6.3. Neuronal-Macrophage Unit

Macrophages such as APDCs related to allergen-presenting cells, but also to phagocytes and efferocytes [124], originate from two sources, including monocytes. All macrophages are divided into two populations, M1 (pro-inflammatary) and M2 (anti-inflammatory), and the M2 population is additionally subdivided into four subsets: M2a (alternatively activated macrophages), M2b, M2c, and M2d (known also as tumor-associated macrophages) [125,126]. Serotonin has been demonstrated to impact upon the differentiation of macrophages from monocytes upregulating the expression of M2-polarization-associated genes and reducing the expression of M1-associated genes [67]. In addition, serotonin inhibits a release of pro-inflammatory cytokines from mature M1 macrophages and via 5-HT_2B_ and 5-HT_2C_ receptors upregulates M2a, M2d, alveolar macrophages, and Kupffer cells [67].

Both M1 and M2 populations express an enormous number of receptors, including receptors for neuro molecules, and secrete many mediators. Tissue-resident macrophages are specialized to the microenvironment and are highly heterogeneous, particularly in the barrier organs.

ACh exerts a dual effect on macrophages of the respiratory tract: when it binds to M3AchR pro-inflammatory effects occur, and, vice versa, via a7nAchR ACh provides the anti-inflammatory activity of macrophages [18]. It has been supposed that local dysfunction of the vegetative nervous system, in particular catecholamines (NE) in the nasal cavity, can be a cause of uncommon sinonasal disorders [127]. 

From the phylogenetic viewpoint, signaling transduction from GABA receptors inside macrophages has been studied in healthy humans and experimental animals of various species, which demonstrated that the human GABA signal transduction is a perfect well-established GABAergic signaling machinery [71]. Therefore, GABA has a potent opportunity to inhibit the macrophage activity.

### 6.4. Neuronal-ILC2Unit

Group 2 innate lymphoid (ILC2) cells located in the subepithelial region are a cell type which responds to so-called epithelium-derived alarmins, IL-25, IL-33, and TSLP, and then secretes a Th2-resembling set of cytokines: IL-4, IL-5, IL-9, IL-13, amphiregulin, and IL-17 [18,60,128]. ILC2 cells are currently an area of research interest because they modulate many immune processes, e.g., neuronal-ILC2 cross-talk promotes mast cell alternative activation pathways [43,60,129]. Recently, some new review articles devoted to the neuronal-ILC2 unit have been published [60,128].

According to some studies, NMU, SP, VIP, and CGRP trigger ILC2 proliferation, then Th2 expansion and expression of allergic inflammation cytokines, including IL-5, though, in most cases, CGRP and VIP exert not pro-inflammatory but anti-inflammatory activities [5,13,17,58,59,83]. Through the cholinergic anti-inflammatory pathway, ACh downregulates ILC2 proliferation [2], and NE limits ILC2-associated type 2 inflammation, and counteracts the activating effects of NMU [63]. According to other publications, CGRP inhibits the activation of ILC2, the Th2-dependent pathway, and manifestation of allergic inflammation [5,82,83]. It turns out that in relation to ILC2, some typically anti-inflammatory neurotransmitters display pro-inflammatory properties, becoming ambivalent [26].

### 6.5. Neuronal-Mast Cell Unit

Mast cells refer to one of two main cellular contributors to the allergic inflammation early phase releasing enormous preformed and newly synthesized mediators [130,131]. Furthermore, serum total mast cell tryptase serves as the “gold standard” biomarker, which can help differentiate anaphylaxis from pseudo anaphylactic shock conditions [132]. The maturation, activation, and functioning of mast cells are regulated by both cytokines and neuro molecules. 

Mast cells enable the synthesis, accumulation, and release of serotonin, but there is no evidence that serotonin modulates their maturation and activation [67]. SP appears to be the most potent neuropeptide activating mast cells and basophils [13], which stimulates their degranulation and the early phase of allergic inflammation wherever it occurs [133]. In addition, SP is able to trigger the alternative pathway of mast cell degranulation through a Mas-related G-protein-coupled receptor—MRGPR [6,54]. L-Glu enables mast cells to cross-talk with neurons of the CNS [40], and the neuronal-mast unit contributes to pain and “neurogenic” inflammation. Neurotrophins such as NGF upregulate the maturation and activation of mast cells [18]. In contrast, adenosine inhibits a release of histamine from mast cells and basophils to diminish inflammatory manifestation [27,89,90].

### 6.6. Neuronal-Eosinophil Unit

Eosinophils, the second type of the most important cells of allergic inflammation, possess primary and specific/crystalloid granules containing IL-5, CCL11 (Eotaxin-1), CCL24 (Eotaxin-2), CCL26 (Eotaxin-3), galectin-10 (Charcot-Leyden crystals), major basic protein, eosinophilic cationic protein, cysteinyl leukotrienes, and other mediators that establish the late phase of allergic inflammation [134,135].

Eosinophil-derived major basic protein operates as an antagonist of M2AchR, inhibiting the negative feedback loop and potentiating oversecretion of ACh that leads to bronchoconstriction in the lungs [22]. L-Glu has been reported to increase in eosinophils upon hyper-responsiveness of the unified airway [41]. Histamine [42] and NMU [59] stimulate eosinophil migration, NGF promotes the activation and degranulation of eosinophils [18], whereas adenosine inhibits their accumulation at the inflammatory sites [89]. Paradoxically, in in vitro experiments serotonin exhibits upregulatory activity in relation to eosinophil migration in humans and mice [67].

### 6.7. Neuronal-Neutrophil Unit

Neutrophils are currently recognized as essential cells taking part in allergic inflammation since the Th2-low/Th17/neutrophilic endotype was established in bronchial asthma [136,137]. Neutrophils contribute to the inflammatory process of many biomolecules, including a release of reactive oxygen species (ROS), metalloproteinases, pro-inflammatory cytokines, chemokines, and even allergen presentation [138].

NE inhibits the activity of neutrophils in the biological fluids, the local environment, receptor expression, and costimulation in a dose-dependent manner [22]. Furthermore, it was recently reported that NE downregulates neutrophil-mediated innate immunity in the lung and skin [5,65,66]. Sensory neuron-derived CGRP was also shown to inhibit neutrophil chemotaxis and phagocytic activity [17,65].

### 6.8. Neuronal-NEC Unit

Each NEC, a cell messenger that links the nervous, endocrine, and immune systems, is commonly located between two epitheliocytes as a rare type of mucociliary barrier non-epithelial cells in the unified airway. In the lung, NEC is termed the “pulmonary NEC” (PNEC), and, in the gut, this cellular lineage is called the “enteroendocrine cell”. Recent scRNA-seq analyses revealed that PNECs account for 0.01% of all lung cells [11]. NECs are often located next to passing neuronal fibers and the location of ILC2, and are proliferated if allergic inflammation occurs [17].

NECs secrete many neuro molecules, such as NMU, CGRP, VIP, NE, serotonin, GABA, and NGF, which affect a number of target immune cells. Neuropeptides NMU, VIP, and CGRP, produced by NEC, activate ILC2 and Th2 cells and stimulate an appropriate set of cytokine synthesis. Particularly, it is noteworthy how IL-5 exerts a gradual increase in the synthesis and engagement of eosinophils to the sites of allergic inflammation [17]. However, there are many controversial data concerning NEC effects on immune cells [17,22,23,82,83]. So far, the neuronal-NEC unit remains a subject of further research and discussion in the literature. 

## 7. Discussion

Sensory neurons of the nervous system and antigen-recognizing receptors of the immune system cells communicate to sense danger. Neurotransmitters and neuropeptides synthesized in neurons and non-neuronal cells have a short lifespan but frequently long-term effects, sharing with the endocrine and immune systems and constituting the mutual tridirectional neuroimmune network [12,13,18,22]. It has long been known that these three body compartments are closely linked and, to a large extent, this communication enhances our understanding of health and disease, including allergic conditions [18]. In relation to the neuroimmune network, a new term,“interactome,” which means the neuroimmune interaction network in its tridirectional entirety, has been proposed [1,15]. Identification of the nature of the interactome in health and its plasticity in disease implies the detailed study of interaction between neuro molecules and immune cells, as well as hormones. Interestingly, in the past, the term “bidirectional neuroimmune network” was preferential in the research community.

Currently, the number of neuropeptides among all neuro molecule types significantly exceeds the number of neurotransmitters [33], whereas new neuropeptides are being continuously identified at present. Grouping of all neuro molecules into predominant pro-immunogenic and protolerogenic profiles according to reference [26] turned out to be controversial, and many molecules should be deemed ambivalent. For example, the main protolerogenic neurotransmitter GABA may upregulate goblet cells’ hyperplasia, promoting inflammatory remodeling of the epithelium in respiratory allergies [74]. The research on cytokines and chemokines is being repeated at a new level, and novel questions are also being generated for the research community as in the past. In particular, an area of interest has been cross-talk between a single cell and influence factors that become accessible for study in the era of the “omics revolution” [100].

Each cell subset within the airway epithelium plays a certain role, and if damaged, may cause functional deviations in the whole cell set prone to pathology [8]. For example, if the mucociliated epithelial barrier integrity is impaired, an allergen or infection may enter the nose and then lung more easily [10,105]. With the aid of scRNA-seq, a real picture of long known and newly revealed cell lineages in the respiratory tract [7,8], in particular the nose [9], have been identified. However, there are almost no publications devoted to the updated cell type repertoire and its cross-talk with immune cells and neuro molecules in allergic rhinitis [6]. Such research relating to local respiratory pathology has not been carried out at all.

In the near future, new generations of transcriptomic and proteomic techniques will use the study of the cell landscape in distinct allergic conditions such as conventional and local allergies to enable us to improve our understanding of cell molecular phenotype and regulatory neuro molecule activity within the nasal epithelium and subepithelial region [7]. Furthermore, it will allow a comparison of these data in canonical and non-canonical disorders with the hope to develop more effective therapeutic approaches for allergic diseases in total and separate individuals when traditional therapy does not work [6]. Failed pharmacotherapy, e.g., with β_2_-agonists, in some patients with allergic rhinitis may be presumably explained by a decreased expression of β_2_AR or local functional disability of catecholamines. Therefore, each medication used in allergic rhinitis has to be analyzed in terms of the impact on neuronal-immune cell units because this may be a hidden cause behind the low efficacy of some drugs. The next approach, which was in part introduced into medical practice, is the use of neuropeptides as medications in allergies [72,73]. According to publications [6,104,105], the most significant neurotransmitters and neuropeptides for the nose in allergic rhinitis have been identified. They are ACh, NE, SP, NMU, VIP, and CGRP, therefore it is these molecules that should be a priority subject of potential pharmacological developments in the near future.

## Figures and Tables

**Figure 1 ijms-23-06938-f001:**
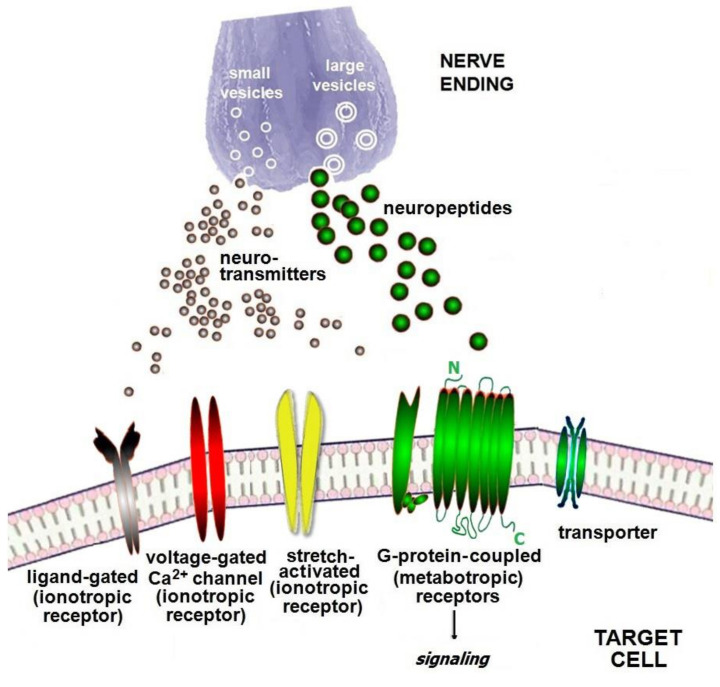
Receptors for neurotransmitters and neuropeptides. The nerve ending enables the release of two types of neuro molecules, neurotransmitters from the small vesicles and neuropeptides from large vesicles. Receptors for neurotransmitters expressed on the target immune cell membrane are divided into ionotropic and metabotropic. The ionotropic receptors of some subtypes possess ion channels, whereas metabotropic receptors, or transmembrane GPCRs, use various signaling pathways. Neuropeptides exploit metabotropic-like GPCPs. Transporters help neuro molecules move through the membranes.

**Figure 2 ijms-23-06938-f002:**
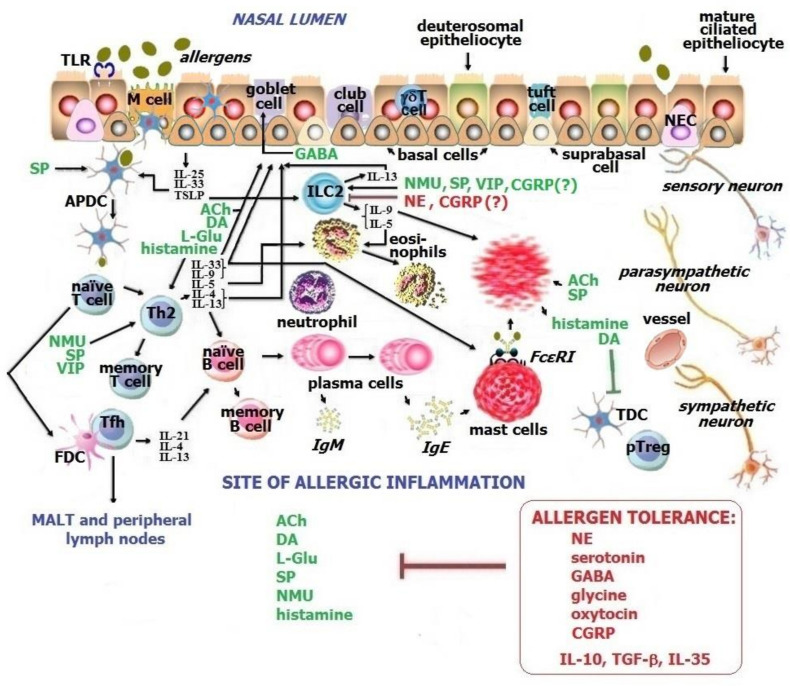
Neuro molecules’ effects in allergic inflammation in the nose. Neurotransmitters and neuropeptides display different effects if allergen tolerance breakdown occurs. ACh and, paradoxically, GABA stimulate the goblet cells to produce mucus. Many neuro molecules, e.g., ACh, DA, L-Glu, histamine, NMU, SP, and VIP, upregulate the Th2-mediated immune response and Th2-associated allergic inflammation. In addition, NMU, SP, and VIP activate ILC2, but NE inhibits the activation of these cells. ACh and SP promote the degranulation of mast cells, DA and histamine downregulate pTregs. CGRP is reported to exert controversial effects. Therefore, pro-immunogenic neuro molecules are predominant in allergic inflammation. ACh—acetylcholine, DA—dopamine, L-Glu—L-glutamate, SP—substance P, NMU—neuromedin U, NE—norepinephrine, GABA—γ Aminobutiric acid, VIP—vasoactive intestinal peptide, CGRP—calcitonin-gene-related peptide, APDC—allergen-presenting dendritic cell, FDC—follicular dendritic cell, TDC—tolerogenic dendritic cell, ILC2—group 2 innate lymphoid cell, Th2—type 2 helper T cell, Tfh—follicular helper T cell, pTreg—peripheral regulatory T cell, M cell—“microfold” cell, TLR—Toll-like receptors, FcεRI—type I IgE receptor, TGF-β—transforming growth factor-β, MALT—mucosae-associated lymphoid tissue. Pro-immunogenic action is noted in green, and protolerogenic action is noted in red.

**Table 1 ijms-23-06938-t001:** The neuroimmune system.

Counterparts	Organs/Tissues	Cells	Particular Phenomena
Neuronal	CNS, spinal cord, and non-neuronal subsystem	Neurons and nerve fibers (somatosensory, viscerosensory, sympathetic, parasympathetic, and enteric) [18,19] Neuroglia Neuron-derived extracellular vesicles [20]	“Neurogenic” inflammation [18,21]
Neuroendocrine	Hypothalamus, pituitary gland, and peripheral endocrine glands (i.e., HPA axis on the whole)	Endocrine gland cells NEC [11,22,23]	General adaptation syndrome [16]
Immune	Primary, secondary, and tertiary organs	Immune cells (lymphocytes and non-lymphoid immune cells) Immune cell-derived extracellular vesicles [24] Barrier epithelium’s cells (ciliated, deuterosomal, goblet, club, basal, suprabasal, tuft cells, ionocytes, etc. [7,10]	Innate immunity Pathways of adaptive immune responses Immune (allergen) tolerance [25,26]

**Table 2 ijms-23-06938-t002:** Neuro molecules: Category, source, and functional activity.

Neuro Molecule	Some Receptors	Category and Prevalent Source	Predominant Activity in Relation to Allergen Tolerance	References
Acetylcholine (ACh)	a7nAchR; M1AchR-M4AchR	Excitatory neurotransmitter (parasympathetic neurons, NEC, lymphocytes, monocytes)	Pro-immunogenic	[18,22,34,35,36,37]
Dopamine (DA)	D_1_-D_5_	“Critical” excitatory and inhibitory neurotransmitter [2] (sympathetic neurons, lymphocytes, dendritic cells (DCs), macrophages, neutrophils)	Pro-immunogenic and ambivalent	[2,27,29,38,39]
L-glutamate (L-Glu)	mGluRs-iGluRs	“Critical” excitatory neurotransmitter [2] (neurons, neuroglia, and other non-neuronal cells)	Pro-immunogenic	[2,27,28,40,41]
Histamine	H_1_-H_4_	Excitatory neurotransmitter (neurons, mast cells, basophils, T cells, enterochromaffin tissue)	Pro-immunogenic	[42,43,44,45,46]
Arginine-vasopressin	AV1AR, AV1BR, AV2R	Neurohormone, or a peptide hormone (the posterior pituitary gland)	Pro-immunogenic	[47,48,49]
Melatonin	MT_1_-MT_3_	Neurohormone, or a peptide hormone (the pineal gland)	Pro-immunogenic	[50,51,52,53]
Substance P (SP)	NK1R-NK3R	“Critical” neuropeptide [2] (sensory neurons, microglia, lymphocytes, DCs, macrophages, eosinophils)	Pro-immunogenic	[5,6,13,27,54,55,56,57]
Neuromedin U (NMU)	NMUR1-NMUR2	Neuropeptide (parasympathetic sensory neurons, NEC)	Pro-immunogenic	[5,6,13,17,22,58,59,60,61,62]
Norepinephrine (NE)	β_2_AR; αAR	Excitatory neurotransmitter (sympathetic neurons, adrenal medulla, lymphocytes, NK cells, monocytes, macrophages, NEC)	Protolerogenic	[22,27,63,64,65,66]
Serotonin (5-hydroxytryptamine)	5-HT_1_-5-HT_7_	“Critical” inhibitory neurotransmitter [2] (central and enteric neurons, enterochromaffin tissue, NEC)	Protolerogenic	[2,27,67,68,69,70]
γ Aminobutyric acid (GABA)	GABA_A_-GABA_B_	Inhibitory neurotransmitter (neurons, T cells, macrophages, NEC)	Protolerogenic	[27,71,72,73,74]
Glycine	GlyRs	Inhibitory neurotransmitter (various neurons)	Protolerogenic	[75,76,77]
Oxytocin	OXTR	Neurohormone, or a peptide hormone (the posterior pituitary gland)	Protolerogenic	[78,79,80,81]
Calcitonin-gene-related peptide (CGRP)	CLRs	Neuropeptide (sensory neurons, T cells, B cells, NEC, the thyroid gland)	Protolerogenic and ambivalent	[3,6,17,18,22,27,60,82,83,84]
Vasoactive intestinal peptide (VIP)	VPAC1-VPAC2	Neuropeptide (parasympathetic sensory neurons, NEC, the gut)	Protolerogenic and ambivalent	[5,6,13,17,18,22,27,60,85,86]
Endocannabinoids	CB1-CB2	Atypical neurotransmitters (various neurons)	Protolerogenic	[27,87,88]
Adenosine ATP	A1-A3; P1, P2X, P2Y	Atypical neurotransmitters (various cells, a DAMP)	Protolerogenic	[27,89,90,91]
Endorphins	μ, δ, κ opioid Rs	Peptides (the pituitary gland, thymus, lymphocytes)	Protolerogenic	[27,92,93,94]
Nerve growth factor (NGF) Brain-derived neurotrophic factor (BDNF) Neurotrophin 3 Neurotrophin 4	TrkA-TrkC; p75NTR	Neurotrophins (sensory neurons, B cells, monocytes, mast cells, eosinophils, keratinocytes, smooth-muscle cells, NEC)	Pro-immunogenic or protolerogenic	[18,32,95,96,97]

## Data Availability

Not applicable.

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
