# Peer review of "Neuronal-Immune Cell Units in Allergic Inflammation in the Nose"

_ijms, 2022, doi:10.3390/ijms23136938_

Round 1
Reviewer 1 Report
I think that the review is very interesting and of high impact for clinical practice, as allergic diseases are becoming more frequent nowadays and there are still treatments that are not effective if they are prolonged over time. For all these reasons I think that this review could be of interest for the readers of the journal, but I have some minor comment for the authors:
It is mentioned in the text that omics technologies could be useful for the study of mechanisms implicated in allergic diseases but then only results from transcriptomics are mentioned. Is there any mass spectrometry study applied to this field? If there is none it should be mentioned in the text.
Author Response
Dear Doctor,
Many thanks for your kind reviewing my manuscript and helpful comment.
I'm submitting the revised manuscript with highlighted addition according to your comment on page 6, including new references (101-103).
Thank again,
Best regards,
Prof. Klimov

Reviewer 2 Report
The paper is well written and of interest. It contains 2 tables and 2 figures and all the paper relevant sections. Its topic is Neuronal-Immune Cell Units in Allergic Inflammation in the 2 Nose. It also contains abbreviations
The authors studied Neuro molecule's effects on allergic inflammation in the nose inc T and B cells neuronal dendrites and mast cells.
There are no major spelling/grammar errors in the paper thus is can be accepted for publication as is.
Author Response
Dear Doctor,
I'd like to express my gratitude to you for the kind reviewing.
Thank you.
Sincerely,
Prof. Klimov